# Environmental predictors of pulmonary nontuberculous mycobacteria (NTM) sputum positivity among persons with cystic fibrosis in the state of Florida

Sydney L. Foote[ID][1], Ettie M. Lipner[2,3], D. Rebecca Prevots[4], Emily E. Ricotta[ID][4]*

1 Office of Data Science and Emerging Technologies, Office of Science Management and Operations, National Institute of Allergy and Infectious Diseases (NIAID), National Institutes of Health (NIH), Rockville, MD, United States of America, 2 Center for Genes, Environment, and Health, National Jewish Health, Denver, CO, United States of America, 3 Department of Epidemiology, Colorado School of Public Health, Aurora, CO, United States of America, 4 Division of Intramural Research, Epidemiology and Population Studies Unit, NIAID, NIH, Rockville, MD, United States of America

* emily.ricotta@nih.gov

**Data Availability Statement:** Data cannot be shared publicly due to data privacy obligations and to maintain the CF Foundation's obligation and commitment to protecting the privacy of people

## Abstract

Nontuberculous mycobacteria (NTM) are opportunistic human pathogens that are commonly found in soil and water, and exposure to these organisms may cause pulmonary nontuberculous mycobacterial disease. Persons with cystic fibrosis (CF) are at high risk for developing pulmonary NTM infections, and studies have shown that prolonged exposure to certain environments can increase the risk of pulmonary NTM. It is therefore important to determine the risk associated with different geographic areas. Using annualized registry data obtained from the Cystic Fibrosis Foundation Patient Registry for 2010 through 2017, we conducted a geospatial analysis of NTM infections among persons with CF in Florida. A Bernoulli model in SaTScan was used to identify clustering of ZIP codes with higher than expected numbers of NTM culture positive individuals. Generalized linear mixed models with a binomial distribution were used to test the association of environmental variables and NTM culture positivity. We identified a significant cluster of *M. abscessus* and predictors of NTM sputum positivity, including annual precipitation and soil mineral levels.

## Introduction

Nontuberculous mycobacteria (NTM) are opportunistic human pathogens that reside in the environment and are commonly found in soil and water [1]. Exposure to these organisms may cause pulmonary nontuberculous mycobacterial (PNTM) disease, which poses a threat to high-risk groups including older adults and individuals with chronic lung conditions; persons with CF are particularly vulnerable [2]. Infection likely results from a combination of behavioral and environmental exposure which jointly increase the risk of NTM infection [3]. Studies have tried to evaluate common exposure sources including household plumbing such as showerheads [4,5], water heating units, and dust, as well as external sources such as soil, watersheds,

with CF who allow their information to be included in the Registry. Data are available from the CF Foundation for researchers who meet the criteria for access to confidential data. Please contact datarequests@cff.org or visit https://www.cff.org/Research/Researcher-Resources/Tools-and-Resources/Patient-Registry-Data-Requests/ for instructions on how to obtain the registry data.

**Funding:** SLF was supported in part by an appointment to the National Institute of Allergy and Infectious Diseases (NIAID) Emerging Leaders in Data Science Research Participation Program. This program is administered by the Oak Ridge Institute for Science and Education through an interagency agreement between the U.S. Department of Energy (DOE) and NIAID. ORISE is managed by ORAU under DOE contract number DE-SC0014664. EML was supported by the Cystic Fibrosis Foundation, Clinical Pilot and Feasibility Award. EER and DRP were supported by the Division of Intramural Research, NIAID, National Institutes of Health. All opinions expressed in this paper are the author's and do not necessarily reflect the policies and views of NIAID, DOE, or ORAU/ORISE.

**Competing interests:** The authors have declared that no competing interests exist.

and climatic factors [6–8]. These factors play a role in determining environmental suitability for the pathogen, with higher abundance of NTM associated with increased rainfall, humidity, certain watersheds, and soil composed of particular elements. Studies have also shown that prolonged exposure to certain environments can increase the risk of PNTM [9]. It is therefore important to determine the risk associated with different geographic areas, so high risk individuals can either avoid an area or focus prevention efforts to reduce their risk of exposure. Previous studies have shown significant clustering of NTM in multiple regions of the United States, including Florida, with geographical heterogeneity in overall and species-specific risk of pulmonary disease. In Florida, the 5-year NTM sputum positivity prevalence was 31% from 2010 through 2014 [2], and this state has the highest prevalence of NTM in the contiguous US [10,11]. Identifying risk factors for exposure and subsequent infection with NTM are central to prevention efforts in the CF community [12]. In this study we describe spatial clusters of NTM infections in Florida and identify environmental predictors of NTM sputum positivity.

## Materials and methods

### Data sources

We conducted a nested case-control study, using annualized registry data obtained from the Cystic Fibrosis Foundation Patient Registry (CFFPR) for 2010 through 2017 [13]. Our study population comprised patients aged $\geq$ 12 years residing $\geq$ 2 consecutive years in Florida. Patients with a history of lung transplant or *Mycobacterium tuberculosis* infection were excluded. Incident NTM cases were persons with a positive pulmonary culture after $\geq$ 1 negative culture(s) and with residence in Florida the year before and the year of their first positive NTM culture. Controls were defined as persons with $\geq$1 negative NTM culture(s) during the study period, residence in Florida the year of and the year before a negative culture, and no positive cultures during Florida residency. Because the majority of individuals had multiple cultures, we used the first culture and residential ZIP code associated with the year of first culture for each person meeting these criteria for analysis.

We selected environmental variables for analysis based on prior findings. Variables that have been previously found to be predictive of sputum positivity include evapotranspiration [10], saturated vapor pressure [14], vapor pressure [15], temperature [6], and rainfall [6], as well as soil or water mineral concentration including copper [10], sodium [10], manganese, [8,10], calcium [7], and molybdenum [7]. Environmental data sources used in this study are described in Table 1. Soil geochemistry collected from 2007 through 2010 included data on calcium, copper, molybdenum, manganese, and sodium content from samples measured in the top 5 cm of soil. Annual temperature and rainfall, and eight-day evapotranspiration data were extracted for the years 2010 through 2017. Evapotranspiration was averaged to create annual estimates. Ordinary kriging was performed to estimate the broader spatial distribution over Florida from the original sampling sites or weather stations for all soil geochemistry variables, temperature, and rainfall. To adapt county-level census data to the ZIP-code level, we determined the county each ZIP code primarily fell within by overlaying ZIP code polygons on a map of Florida counties using the R packages "rgdal" and "sp" [16–18]. The ZIP code-level mean of each environmental variable was calculated, scaled, and mean-centered by year, when data were available, for analysis. Driving distance from ZIP code centroids to CF clinics in Florida were calculated using a Distance Matrix API via Google Cloud Services and the R package "gmapsdistance" [19] to control for potential spatial clustering near CF clinics. The closest clinic for individuals was assumed to be the pediatric or adult clinic with the shortest driving distance for all patients in each ZIP code under or at least 18 years of age, respectively.

**Table 1. Data sources for variables used in analysis.**

| Variable | Source | Type and frequency | Period | Citation |
|---|---|---|---|---|
| Soil minerals | United States Geological Services' (USGS) Geochemical and Mineralogical Data for Soils of the Conterminous United States study | Survey sites, collected once | 2007–2010 | https://pubs.usgs.gov/ds/801/ |
| Temperature | National Oceanic and Atmospheric Administration (NOAA) Global Summary of the Year (GSOY) database | Weather station, annual | 2010–2017 | https://data.nodc.noaa.gov/cgi-bin/iso?id=gov.noaa.ncdc:C00947 |
| Rainfall | National Oceanic and Atmospheric Administration (NOAA) Global Summary of the Year (GSOY) database | Weather station, annual | 2010–2017 | https://data.nodc.noaa.gov/cgi-bin/iso?id=gov.noaa.ncdc:C00947 |
| Evapotranspiration | MOD16A2 Version 6 Evapotranspiration/Latent Heat Flux dataset by the National Aeronautics and Space Administration (NASA) in collaboration with the USGS | Remotely sensed, 8-day | 2010–2017 | https://lpdaac.usgs.gov/products/mod16a2v006/ |
| CFF centers | Cystic Fibrosis Foundation website | Not applicable | | https://www.cff.org/ccd/CareCenters?State=FL&Zip=&Distance=100 |
| Median income | United States Census Bureau American Community Survey (ACS) data | Survey, annual | 2010–2017 | https://www.census.gov/programs-surveys/acs/data.html |
| Metropolitan centers | Centers for Disease Control and Prevention (CDC) 2013 National Center for Health Statistics (NCHS) Urban-Rural Classification Scheme for Counties | Census-based data, collected once | 2013 | https://www.cdc.gov/nchs/data_access/urban_rural.htm |

## Spatial and statistical analysis

We used a Bernoulli model comparing individuals with CF who were NTM culture positive (cases) to individuals with CF who were NTM culture negative (controls) to identify clustering of ZIP codes with higher than expected numbers of NTM culture positive individuals from 2011 through 2017, using SaTScan version 9.6 [20] with default setting, limiting cluster radius to 50km. This was done as a way to assess the presence of geographic locations in Florida where people with CF might be at increased risk of NTM. We repeated the analysis for overall NTM as well as each species of NTM separately. To test the association of environmental variables and NTM culture positivity, we used generalized linear mixed models with a binomial distribution. The variance inflation factor was used to assess collinearity and model fit was assessed via Akaike Information Criterion. The final environmental variables included were copper, manganese, molybdenum, sodium, annual precipitation, and evapotranspiration from the year in which an individual had their first NTM culture. Additionally, we included patient gender, age, number of years receiving chronic macrolides through the year of culture, the closest CF clinic in miles, whether the ZIP code was considered metropolitan, ZIP code median household income, and a random intercept for ZIP code. Statistical analyses were conducted using R versions 3.6.1–4.0.2 [21]. The study was determined to be not Human Subjects Research by the NIH Office of Human Subjects Research Protection.

## Results

Of the 1293 patients in the CFFPR residing ≥ 2 consecutive years in Florida from 2010 through 2017, 979 patients met inclusion criteria; 261 (26.7%) were classified as cases and 718 (73.3%) as controls (Fig 1). Species identified were distributed as follows: 109 (41.8%) *M. avium* complex (MAC), 118 (45.2%) *M. abscessus* and its subspecies, and 62 (23.8%) other species. The proportions of sex, age, and years of chronic macrolide use were similar between cases and controls, while years of Florida residency from 2010 to the year of the first culture were generally greater for cases (Table 2). We found one statistically significant high-risk cluster in southeast Florida (p-value: 0.035, radius: 48.8 km), which included Broward, Miami-

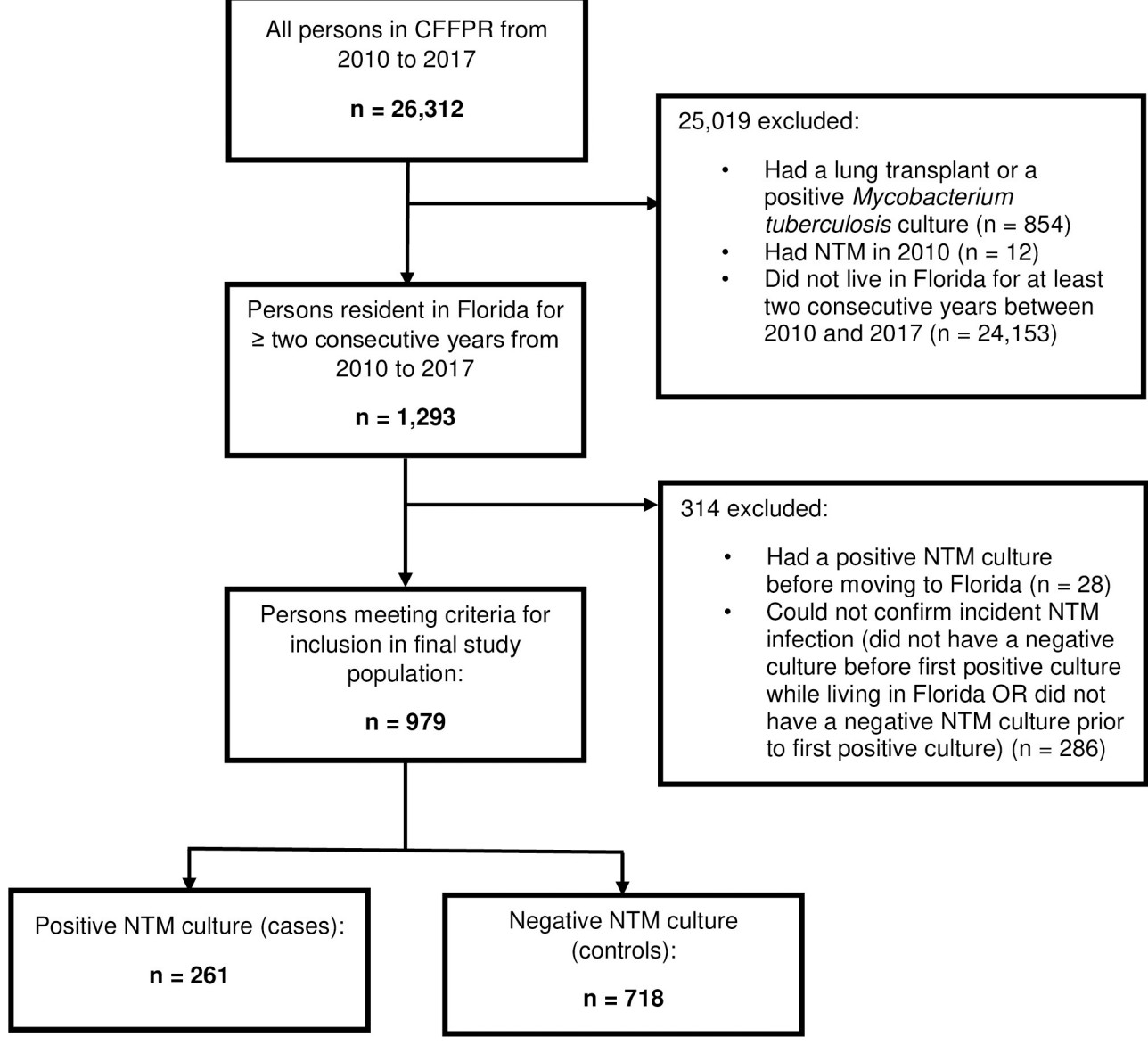

**Fig 1. Study population for incident NTM cases and negative controls in the CFFPR from 2010 through 2017.**

Dade, and Palm Beach counties (Fig 2). This high risk-cluster was associated with *M. abscessus*; no significant clustering was observed for other species.

Sputum-positive patients were more likely to live in a ZIP code with higher average yearly precipitation, with a 34% increase in the odds of a positive culture for each standard deviation (SD) increase in average annual precipitation (adjusted odds ratio [aOR]: 1.34, 95% confidence interval [95% CI]: 1.13–1.58). Soil geochemistry was also associated with NTM positivity; a one SD increase in levels of sodium in the soil was associated with a 92% increased risk of culture positivity (aOR: 1.92, 95% CI: 1.46–2.52), and a one SD increase in soil manganese was associated with a 40.7% decreased risk (aOR: 0.59, 95% CI, 0.46–0.77). Species-specific analysis showed the same associations for *M. abscessus* (Table 3).

**Table 2. Baseline demographics of the study population by NTM culture result, Florida, 2011–2017.**

| | NTM cases* (n = 261) | | | Controls (n = 718) |
|---|---|---|---|---|
| | **All NTM (n = 261)** | **MAC* (n = 109)** | **Mabs* (n = 118)** | |
| Sex | | | | |
| Female | 119 (45.6) | 51 (46.8) | 52 (44.1) | 361 (50.3) |
| Male | 142 (54.4) | 58 (53.2) | 66 (55.9) | 357 (49.7) |
| Age group, yr† | | | | |
| 12 to <18 | 80 (30.7) | 38 (34.9) | 44 (37.3) | 251 (35.0) |
| 18 to <60 | 178 (68.2) | 69 (63.3) | 73 (61.9) | 462 (64.3) |
| ≥ 60 | 3 (1.1) | 2 (1.8) | 1 (0.8) | 5 (0.7) |
| Chronic macrolide use‡ | | | | |
| 0 yr | 68 (26.1) | 25 (22.9) | 40 (33.9) | 195 (27.2) |
| 1–2 yr | 88 (33.7) | 43 (39.4) | 49 (41.5) | 332 (46.2) |
| 3–4 yr | 66 (25.3) | 24 (22.0) | 21 (17.8) | 122 (17.0) |
| 5+ yr | 39 (14.9) | 17 (15.6) | 8 (6.8) | 69 (9.6) |
| Florida residency‡ | | | | |
| 2–3 yr | 129 (49.4) | 58 (53.2) | 69 (58.5) | 620 (86.4) |
| 4–5 yr | 80 (30.7) | 27 (24.8) | 31 (26.3) | 71 (9.9) |
| 6+ yr | 52 (19.9) | 24 (22.0) | 18 (15.3) | 27 (3.8) |

*NTM = nontuberculous mycobacteria; MAC = Mycobacterium avium complex; Mabs = *M. abscessus*.

†Age groups are determined by year of NTM culture.

‡From 2010 to year of NTM culture.

## Discussion

Because the prevalence of PNTM in Florida is so high [10,11], understanding whether there are environmental predictors or geospatial clustering of NTM is of interest to the CF community and public health. We found a significant cluster of NTM culture positivity, specifically *M. abscessus*, among persons with CF living in Florida in the southeast part of the state. *M. abscessus* is one of the most commonly isolated species of NTM from persons with CF, found in up to 16–68% of NTM-positive sputum cultures [22,23] and is considered difficult to treat due to high levels of inducible resistance to macrolides, the typical first-line therapy for infections with other NTM species [24]. It is therefore important to understand geographical areas of higher risk to acquiring this pathogen.

In addition to spatial clustering, we also found an association between sputum positivity and annual precipitation, soil sodium levels, and levels of soil manganese. The risk associated with higher sodium and lower manganese levels in soil is consistent with two other studies in the US, however these studies were not able to examine species-specific associations [8,10]. An association with rainfall has been recently identified for the province of Queensland, Australia; this relationship varied by species and geographic region [6]. One of the challenges of studying the environmental risks for PNTM disease is that the incubation period for PNTM is unknown, which creates uncertainty about the appropriate timescale for measuring exposure. While we studied incident infections to limit this potential bias, it will be important to conduct further analyses varying the timescale of possible exposure to environmental variables. Additionally, longitudinal, granular data on soil geochemistry is currently unavailable; the soil mineral concentrations throughout Florida likely vary more than represented in our study, limiting our ability to adjust these variables based on time and location. These are particularly important considerations as studies have shown that prolonged exposure to certain

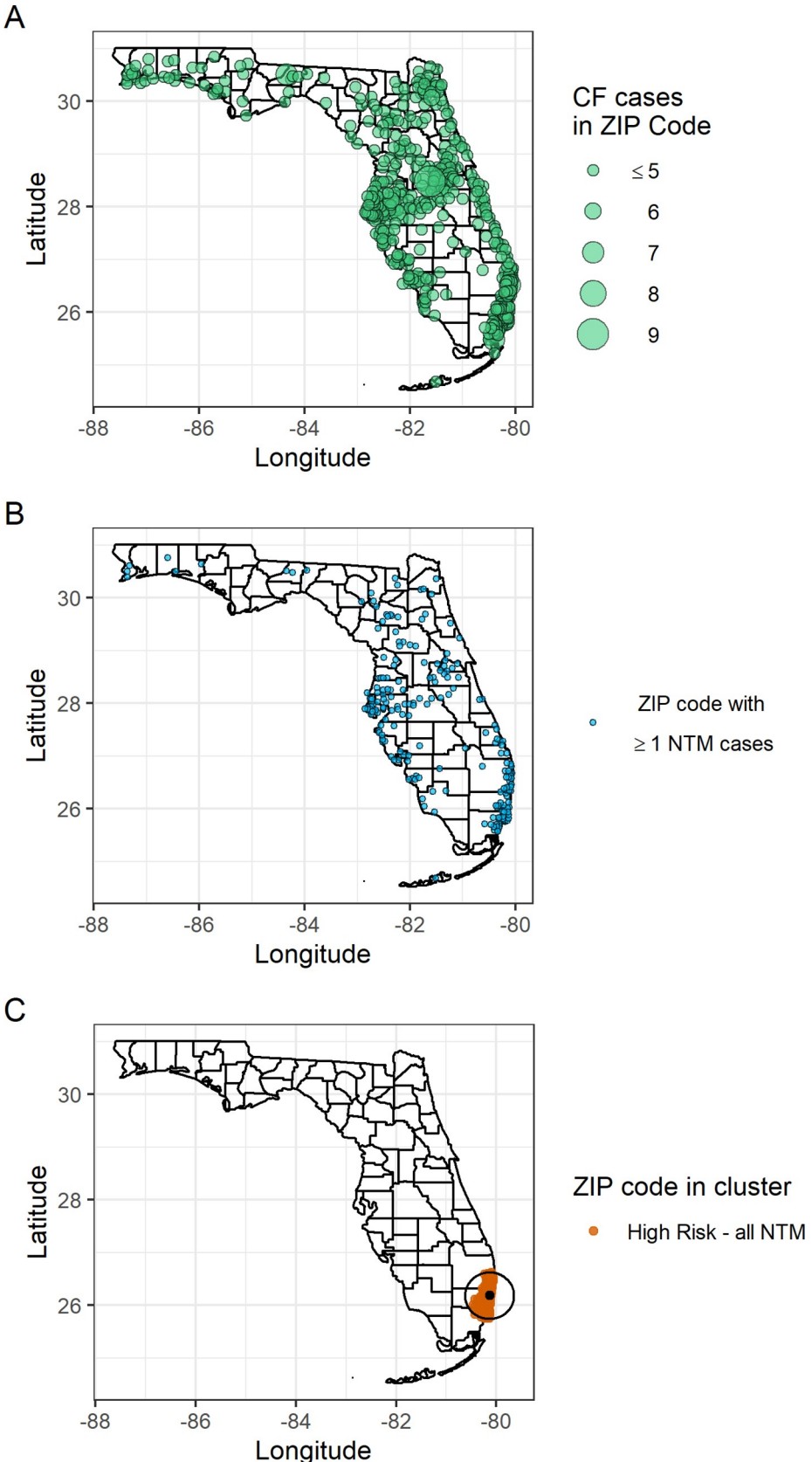

**Fig 2. Distributions of persons residing in Florida for ≥ 2 years from 2011 through 2017. A.** Persons with cystic fibrosis (CF), n = 979. **B.** Nontuberculous mycobacteria (NTM) cases among persons with CF, n = 261. **C.** ZIP codes within a high-risk cluster of NTM cases among persons with CF.

**Table 3. Generalized linear mixed models with a binomial distribution and a random intercept for zip code of demographic and environmental risk factors for NTM culture positivity in persons with cystic fibrosis, Florida, 2011–2017.**

| | | | All NTM* | MAC* | Mabs* |
|---|---|---|---|---|---|
| | Fixed effects | SD* | aOR (95% CI)* | aOR (95% CI) | aOR (95% CI) |
| Individual level | Female | - | 0.81 (0.60, 1.09) | 0.91 (0.59, 1.41) | 0.75 (0.50, 1.12) |
| | Age, per year | - | 1.01 (1.00, 1.03) | 1.00 (0.98, 1.02) | 1.01 (0.99, 1.03) |
| | Years on chronic macrolide | - | 1.10 (1.02, 1.19) | 1.10 (0.98, 1.23) | 0.88 (0.79, 0.99) |
| Zip code level | Mean precipitation† | 7.4 in | 1.34 (1.13, 1.58) | 1.18 (0.92, 1.51) | 1.25 (1.00, 1.57) |
| | Mean yearly evapotranspiration† | 895 mm | 0.85 (0.70, 1.04) | 0.75 (0.54, 1.03) | 1.03 (0.78, 1.36) |
| | Mean soil manganese† | 16.8 ppm | 0.59 (0.46, 0.77) | 0.82 (0.58, 1.17) | 0.53 (0.34, 0.82) |
| | Mean soil sodium† | 46.2 ppm | 1.92 (1.46, 2.52) | 1.44 (0.98, 2.13) | 2.27 (1.45, 3.54) |
| | Closest CF clinic*† | 32.5 miles | 0.90 (0.76, 1.07) | 0.76 (0.57, 1.02) | 0.97 (0.76, 1.25) |
| | Median household income, per $1000† | $17.10 | 0.98 (0.84, 1.15) | 1.01 (0.80, 1.28) | 1.17 (0.96, 1.42) |
| | In a metropolitan area | - | 0.76 (0.37, 1.56) | 0.72 (0.25, 2.07) | 0.68 (0.24, 1.93) |

*NTM = nontuberculous mycobacteria; MAC = Mycobacterium avium complex; Mabs = *M. abscessus*; CF = cystic fibrosis; SD = standard deviation; aOR (95% CI) = adjusted odds ratio (95% confidence interval).

†Variables were scaled and centered to zero; interpretation is by standard deviation.

environments increases the risk of PNTM [9], so cumulative exposures to a variety of high risk sources may increase the risk of infection.

Climatic and environmental factors that contribute to increased mycobacterial abundance likely vary by region, making identification of a uniform set of determinants contributing to PNTM disease across national and global geographic areas challenging. Factors related to the built environment also likely interact with soil and water sources to affect the presence of mycobacteria in the environment; a recent study quantifying mycobacteria in showerheads found that both type of chlorination and showerhead material influenced their abundance [5]. Future studies which estimate risk related to mycobacterial abundance as well as components of the natural and built environmental will allow more complete and precise elucidation of these factors. Because persons with CF are at increased risk for NTM infection, continued studies to determine high-risk geographical areas and specific predictors of disease are critical so that precautions can be taken to reduce risk of exposure.

## Acknowledgments

The authors thank the patients, care providers, and clinic coordinators at CF centers throughout the United States for their contributions to the Cystic Fibrosis Foundation Patient Registry.

## Author Contributions

**Conceptualization:** D. Rebecca Prevots, Emily E. Ricotta.

**Data curation:** Sydney L. Foote, Emily E. Ricotta.

**Formal analysis:** Sydney L. Foote, Emily E. Ricotta.

**Funding acquisition:** D. Rebecca Prevots.

**Investigation:** Emily E. Ricotta.

**Methodology:** Sydney L. Foote, Ettie M. Lipner, D. Rebecca Prevots, Emily E. Ricotta.

**Project administration:** Emily E. Ricotta.

**Supervision:** D. Rebecca Prevots, Emily E. Ricotta.

**Writing – original draft:** Sydney L. Foote, Ettie M. Lipner, D. Rebecca Prevots, Emily E. Ricotta.

**Writing – review & editing:** Sydney L. Foote, Ettie M. Lipner, D. Rebecca Prevots, Emily E. Ricotta.

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
