## [Decision Letter · Decision Letter 0]

20 Apr 2021

PONE-D-21-06131

Environmental predictors of pulmonary nontuberculous mycobacteria (NTM) sputum positivity among persons with cystic fibrosis in the state of Florida

PLOS ONE

Dear Dr. Ricotta,

Thank you for submitting your manuscript to PLOS ONE. After careful consideration, we feel that it has merit but does not fully meet PLOS ONE’s publication criteria as it currently stands. Therefore, we invite you to submit a revised version of the manuscript that addresses the points raised during the review process.

Both reviewers found your study to be a potentially important contribution to the epidemiology of NTM infections in CF patients based on geographic location. If you choose to submit a revised manuscript, Reivew 2 has a number of comments all of which need to be addressed in your response, along with revision of the manuiscript.

We look forward to receiving your revised manuscript.

Kind regards,

Thomas Byrd

Academic Editor

PLOS ONE

Journal Requirements:

Additional Editor Comments:

Both reviewers found your manuscript interesting and a potentially important contribution to the role that geographic location plays in the incidence of NTM lung infections in CF patients. Reivewer 2 raises a number of questions that need to be addressed. If you choose to submit a revised version of your manuscript, please respond to the issues raised by Reviewer 2 and modify the manuscript accordingly.

Reviewers' comments:

Reviewer's Responses to Questions

**Comments to the Author**

1. Is the manuscript technically sound, and do the data support the conclusions?

Reviewer #1: Yes

Reviewer #2: No

2. Has the statistical analysis been performed appropriately and rigorously? 

Reviewer #1: Yes

Reviewer #2: No

3. Have the authors made all data underlying the findings in their manuscript fully available?

Reviewer #1: Yes

Reviewer #2: Yes

4. Is the manuscript presented in an intelligible fashion and written in standard English?

Reviewer #1: Yes

Reviewer #2: Yes

5. Review Comments to the Author

Reviewer #1: An important contribution toward understanding the environmental characteristics that are determinants of environmental mycobacterial geographical distribution.

I read the article closely, but could find no instances where I thought changes or additions/deletions were needed.

Reviewer #2: Review: Foote et al. Environmental predictors of pulmonary nontuberculous mycobacteria (NTM) sputum positivity among persons with cystic fibrosis in the state of Florida

Manuscript (MS) in review. Interesting and much needed analysis of environmental risk factors for NTM sputum culture positivity. The use of CF patients is a novel and useful approach to identifying those with substantial exposure to NTM. The use of cluster analysis with environmental analysis at the zip code level is a specific and granular approach to identifying potentially important exposures.

General comment. This reader had to reread the manuscript multiple times to understand the role of the SE cluster of NTM cases. Were they modelled separately? It does not appear so, the model encompasses the state. I have come to the conclusion that you called them out because you identify them as a cluster and that is all. This is confusing. Maybe there should be less emphasis on this discovery (no inclusion in Figure 2) or somehow otherwise alert the reader that this was an incidental finding. Did you run a cluster analysis on controls? Maybe looking at case and control clusters is another environmental analysis report (suggestion). But in this report, the cluster is a distraction as currently presented.

Methods

Q- How was zipcode assigned to patient and control? Did you use residential or hospital zipcode or something else? Was the same method used for both patient and control?

Q- You analyze sputum positive CF patients (cases) and compare to controls: What is the rate of false negatives among sputum samples for NTM? Did you evaluate later cultures of controls to see if they became cases later during your six year study period?

Q-Line 64- you analyzed weather, soil or water mineral concentrations. I did not see the resulting values for these environmental exposures (maybe in a supplement)? Nor do I see a data source for water minerals in Table 1, although soil is listed.

Q-How were zip code and census polygons (income, metropolitan centers) reconciled?

The reader needs more information about the temporal and spatial variability of covariates to be comfortable with the presence of environmental exposures included in the final model. Because variables such as temperature, precipitation, and evapotranspiration (“weather”) vary greatly month to month, year to year. Sea surface temperatures influence air temperature and precipitation both influence evapotranspiration. Early in your study period, there were anomalous lows, then anomalous highs, then lows again, ending in anomalous highs in 2017 in sea surface temperatures. See: https://twitter.com/MichaelRLowry/status/1249061412896464896/photo/1

(Questions to authors in bold text) This reader is concerned that there is a lack of information about this variability and how “weather” data are related to NTM culture positivity. What is the typical lag between NTM colonization and detection vs sputum sampling? How often are CF patients cultured? Please cite sources. This reader is seeing that analyses were conducted over a six year period (2011-2017). All subjects appear to be lumped into that time period. Mean “weather” variables are defined by year. What is the distribution of NTM culture positivity by month? Please include table. How is weather associated with sputum positivity over time, given potential lags? Was there enough power to perform a temporal analysis between weather and sputum positivity?

In short, there is a major mismatch between exposure and outcome. If you wish to include exposure variables that can change so much year to year, as I reader I expect to see time in the model. It is Florida. What about outlier precipitation years due to hurricanes? Effect on sputum positivity? I also expect to see the outcome analyzed by time. But NTM is considered a chronic condition. How do you reconcile this analysis of highly temporally variable “weather” with an outcome (sputum positivity) with no granularity or knowledge of time?

Soil is the also problematic. In the absence of additional information in the MS, these are the issues.

1-The reference for soil mineral data: https://pubs.usgs.gov/ds/801/ leads one to a summary report: “U.S. Department of the Interior, U.S. Geological Survey, Data Series 801, Geochemical and Mineralogical Data for Soils of the Conterminous United States” (“Data for Soils”). The report describes sampling methods which include a single point sample collection from “target sites that represented a density of approximately 1 site per 1,600 km2”.

Spatial variability of soil minerals is a large concern for this report. Lines 70-72 state ” Where necessary, ordinary kriging was performed to estimate the broader spatial distribution over Florida from the original sampling sites or weather stations. “ This statement comes after “weather” variables were discussed. Is this kriging approach also true for point estimates of minerals in soil by zipcode?

Please include citations for why a soil mineral analysis of this type -using kriging among spatial points 1,600 km2 apart, sampled once near the time of the study period, represents the exposure of people living among the points during the study period.

When one looks at variability in soil characteristics, this reader was surprised by the extreme temporal and spatial variability in concentrations of minerals. Let’s examine the two that made it to the final model, sodium and manganese.

Sodium: the exposure analysis using grid spacing for sodium samples of 1 site per 1,600 km2 appears to be too coarse to estimate your subjects exposures at the zipcode level. See: Trangmar et al. Application of Geostatistics to Spatial Studies of Soil Properties. Advances in Agronomy 1986. https://doi.org/10.1016/S0065-2113(08)60673-2

Manganese: in a study of agricultural fields, manganese concentrations differed among months of the year and over the extent of the field. Areas with higher concentrations did not keep rank order of concentrations over the year. See: Hoskinson, et al. Temporal Changes in the Spatial Variability of Soil Nutrients. INEEL/CON-99-00290. 1999.

https://www.osti.gov/servlets/purl/9823-CNgTWm/webviewable/

It appears that soil minerals are variable spatially and temporally due to multiple factors such as land use, vegetative cover, soil microbial communities, and precipitation.

Therefore, I fear that your assignment of soil mineral exposure was too simplistic. Please prove me wrong, because we need to better understand environmental risk factors for NTM colonization.

Note- attachment preserves figure and bold text features of review.

6. PLOS authors have the option to publish the peer review history of their article (what does this mean?). If published, this will include your full peer review and any attached files.

Reviewer #1: No

Reviewer #2: No

---

## [Author Response · Author response to Decision Letter 0]

20 Jul 2021

PONE-D-21-06131 - Environmental predictors of pulmonary nontuberculous mycobacteria (NTM) sputum positivity among persons with cystic fibrosis in the state of Florida

Response to reviewers

 Reviewer #2: Review: Foote et al. Environmental predictors of pulmonary nontuberculous mycobacteria (NTM) sputum positivity among persons with cystic fibrosis in the state of Florida

Manuscript (MS) in review. Interesting and much needed analysis of environmental risk factors for NTM sputum culture positivity. The use of CF patients is a novel and useful approach to identifying those with substantial exposure to NTM. The use of cluster analysis with environmental analysis at the zip code level is a specific and granular approach to identifying potentially important exposures.

QUESTION 1: General comment. This reader had to reread the manuscript multiple times to understand the role of the SE cluster of NTM cases. Were they modelled separately? It does not appear so, the model encompasses the state. I have come to the conclusion that you called them out because you identify them as a cluster and that is all. This is confusing. Maybe there should be less emphasis on this discovery (no inclusion in Figure 2) or somehow otherwise alert the reader that this was an incidental finding. Did you run a cluster analysis on controls? Maybe looking at case and control clusters is another environmental analysis report (suggestion). But in this report, the cluster is a distraction as currently presented.

RESPONSE: For clustering analysis, individuals with CF who were NTM culture positive (cases) were compared to individuals with CF who were NTM culture negative (controls) using a Bernoulli model. This analysis was repeated for all NTM cases as well as each species of NTM separately, resulting in finding the cluster of M. abscessus. We performed this analysis to highlight a specific geographical area of Florida where there may be a higher risk of acquiring NTM, important knowledge for persons with CF in addition to the environmental findings of our study. For this type of analysis, controls serve to determine the expected rate of cases in a given area, therefore we would not do a separate analysis on controls because they serve as our comparison group. 

We have clarified the methods and purpose of this type of analysis in the text, which now reads: “We used a Bernoulli model comparing individuals with CF who were NTM culture positive (cases) to individuals with CF who were NTM culture negative (controls) to identify clustering of ZIP codes with higher than expected numbers of NTM culture positive individuals from 2011 through 2017, using SaTScan version 9.6 [17] with default setting, limiting cluster radius to 50km. This was done as a way to assess the presence of geographic locations in Florida where people with CF might be at increased risk of NTM. We repeated the analysis for overall NTM as well as each species of NTM separately."

Methods

QUESTION 2: How was zipcode assigned to patient and control? Did you use residential or hospital zipcode or something else? Was the same method used for both patient and control?

RESPONSE: Residential ZIP codes were obtained from the Cystic Fibrosis Foundation Patient Registry (annualized registry data) for both for both cases and controls. The ZIP codes used were those associated with the year the patients had their first positive culture and for controls, the year they had their first negative culture with Florida residency (i.e., at least two years living in Florida) 

This approach has been clarified in the manuscript: “The first culture and residential ZIP code associated with the year of first culture for each person meeting these criteria was used for analysis.”

QUESTION 3: You analyze sputum positive CF patients (cases) and compare to controls: What is the rate of false negatives among sputum samples for NTM? Did you evaluate later cultures of controls to see if they became cases later during your six year study period?

RESPONSE: The choice of a control group was based on the approach that controls had to be at risk of infection in Florida – in other words, persons with CF that lived in the same state and during the same time period as cases. If any individuals were misclassified as controls but were sputum positive, this would only dilute the observed association between the exposures and the outcome of sputum positivity. For this reason, any disease misclassification resulting from a false negative within our control group would be biasing our findings towards a null association rather than a bias in the opposite direction. 

We are not aware of systematic evaluation of rates of false negatives for nontuberculous mycobacterial cultures among persons with CF. Even though different laboratory techniques could have variable sensitivities, the laboratory methods for NTM species identification are relatively standard. We are not sure if the concern of the reviewer is false negatives related to laboratory techniques or sputum sampling techniques, or other sources of variability in mycobacterial detection. However, there is no reason to believe that the rate of misclassification would vary by zip code and therefore the cohort should be affected homogeneously.

To address the last point, it is possible that individuals may become reinfected or relapse, or that individuals who are negative in one year could become positive later, and individuals who are positive could clear infection. For this reason, we evaluated culture status of controls at follow up. Among controls who remained resident in Florida, none became cases during the six-year follow-up period. Four controls had a positive culture after moving out of Florida; we kept these four controls in our analysis to give our findings more power. 

QUESTION 4: Line 64- you analyzed weather, soil or water mineral concentrations. I did not see the resulting values for these environmental exposures (maybe in a supplement)? Nor do I see a data source for water minerals in Table 1, although soil is listed.

RESPONSE: The variables listed on line 64 are examples of environmental variables previously found to be predictive of sputum positivity. The resulting values for environmental exposures used in analysis can be found in Table 3. We did not analyze water mineral variables in this study. We recognize that the wording of this section may be misleading, so have revised it to:

“We selected environmental variables for analysis based on prior findings. Variables that have been previously found to be predictive of sputum positivity include evapotranspiration [10], saturated vapor pressure [14], vapor pressure [15], temperature [6], and rainfall [6], as well as soil or water mineral concentration including copper [10], sodium [10], manganese, [8,10], calcium [7], and molybdenum [7]. Environmental data sources used in this study are described in Table 1.”

QUESTION 5: How were zip code and census polygons (income, metropolitan centers) reconciled?

RESPONSE: ZIP code and census polygons were reconciled by mapping the census data by county, then overlaying the ZIP code polygons to determine the counties each ZIP codes primarily fell within. We then joined the relating census data to the ZIP codes within each county for further analysis. 

This methodology has been clarified in the manuscript: “To adapt county-level census data to the ZIP-code level, we determined the county each ZIP code primarily fell within by overlaying ZIP code polygons on a map of Florida counties using the R packages “rgdal” and “sp”.” 

The reader needs more information about the temporal and spatial variability of covariates to be comfortable with the presence of environmental exposures included in the final model. Because variables such as temperature, precipitation, and evapotranspiration (“weather”) vary greatly month to month, year to year. Sea surface temperatures influence air temperature and precipitation both influence evapotranspiration. Early in your study period, there were anomalous lows, then anomalous highs, then lows again, ending in anomalous highs in 2017 in sea surface temperatures. See: https://twitter.com/MichaelRLowry/status/1249061412896464896/photo/1

RESPONSE: The authors recognize that there is temporal variability of the covariates that is not addressed in this analysis, however this is for two reasons: first, we only have annualized data on culture positivity for this analysis and second, the incubation period for NTM is not known but could range from months to years. For these reasons, we are unable to match more granular temporal data to positivity data. 

QUESTION 6: This reader is concerned that there is a lack of information about this variability and how “weather” data are related to NTM culture positivity. What is the typical lag between NTM colonization and detection vs sputum sampling? How often are CF patients cultured? Please cite sources. This reader is seeing that analyses were conducted over a six year period (2011-2017). All subjects appear to be lumped into that time period. Mean “weather” variables are defined by year. What is the distribution of NTM culture positivity by month? Please include table. How is weather associated with sputum positivity over time, given potential lags? Was there enough power to perform a temporal analysis between weather and sputum positivity?

RESPONSE: As stated above, the incubation period for NTM is unknown, but is thought to be on the order of month to years, although likely shorter for persons with CF. In persons without CF, the time between symptom onset and diagnosis has been reported to range from months to years (Ratnatunga et al., 2020). Given the unknown incubation period and the delays in diagnosis, it seems reasonable to study associations on the order of years. While it is true that weather is highly variable, on average there is likely to be more variability across regions than within a given region. Although we recognize the limitations of these kinds of analyses, especially given the uncertain incubation period, the fact that our findings are in line with other studies evaluating the association of NTM culture positivity and environmental variables would suggest that these factors do play a role in NTM infection. 

We did not assess the study as a six-year time period. To account for time and the temporal association of positivity with environmental variables, for each individual we used the environmental data associated with the year of the person’s first culture (first positive for cases, first negative for controls). So for example if someone was positive in 2013, we would use the annual mean precipitation, soil variables, and evapotranspiration from 2013. This has been clarified in the text, which now reads: “The final environmental variables included were copper, manganese, molybdenum, sodium, annual precipitation, and evapotranspiration from the year in which an individual had their first NTM culture.” 

We did try lagging the environmental variables (evapotranspiration and temperature, specifically) by a year, but there were no significant effects in this analysis, so they were not included in any of the final models.

Ratnatunga, C. N., Lutzky, V. P., Kupz, A., Doolan, D. L., Reid, D. W., Field, M., Bell, S. C., Thomson, R. M., & Miles, J. J. (2020). The Rise of Non-Tuberculosis Mycobacterial Lung Disease. In Frontiers in Immunology (Vol. 11, p. 303). Frontiers Media S.A. https://doi.org/10.3389/fimmu.2020.00303

QUESTION 7: In short, there is a major mismatch between exposure and outcome. If you wish to include exposure variables that can change so much year to year, as I reader I expect to see time in the model. It is Florida. What about outlier precipitation years due to hurricanes? Effect on sputum positivity? I also expect to see the outcome analyzed by time. But NTM is considered a chronic condition. How do you reconcile this analysis of highly temporally variable “weather” with an outcome (sputum positivity) with no granularity or knowledge of time?

RESPONSE: We agree with the reviewer that weather conditions may be highly variable by time, including by month and year, and that the incubation period, the interval between exposure and infection is unknown. For these reasons, it is challenging to precisely associate exposure with outcome. This limitation is inherent to the multiple “ecologic” environmental studies that have been conducted to date for NTM where we are associating climatic variables, such as precipitation and humidity, at a broad scale with individual disease/infection or clusters of disease without knowing the exact period when a person was exposed. These types of biases would likely lead to misclassification of exposure resulting in associations closer to the null value. The finding in prior studies of associations between climatic factors and sputum positivity supports that these are real associations; for example, a prior national study of sputum positivity among persons with CF used climate data averaged over an 11-year period and found similar associations. This averaging likely smooths out the extreme values [see papers by Adjemian et al in main text references], although it is beyond the scope of the current analysis to be able to say whether an extreme event is a risk factor. Moreover, there is likely more variability across geographic areas within Florida than within areas, such that even an extreme event would likely not change the association a great deal. 

In these particular models, the most precise approach we could take was to associate the temperature, precipitation, and evapotranspiration for each case and control from the year of the first culture that met the study definition. Thus, the model does account for outlier years, such as precipitation due to hurricanes. The relation of time in our model has been clarified throughout the methods.

We reconciled the analysis of climatic variables with sputum positivity by including only incident cases that had at least one negative culture before the first positive culture. However, since the incubation period of pulmonary NTM is unknown, there are unresolvable uncertainties around the appropriate timescales for measuring environmental exposures. This aspect is emphasized as a limitation of our study.

Soil is the also problematic. In the absence of additional information in the MS, these are the issues.

1-The reference for soil mineral data: https://pubs.usgs.gov/ds/801/ leads one to a summary report: “U.S. Department of the Interior, U.S. Geological Survey, Data Series 801, Geochemical and Mineralogical Data for Soils of the Conterminous United States” (“Data for Soils”). The report describes sampling methods which include a single point sample collection from “target sites that represented a density of approximately 1 site per 1,600 km2”.

QUESTION 8: Spatial variability of soil minerals is a large concern for this report. Lines 70-72 state ” Where necessary, ordinary kriging was performed to estimate the broader spatial distribution over Florida from the original sampling sites or weather stations. “ This statement comes after “weather” variables were discussed. Is this kriging approach also true for point estimates of minerals in soil by zipcode?

Please include citations for why a soil mineral analysis of this type -using kriging among spatial points 1,600 km2 apart, sampled once near the time of the study period, represents the exposure of people living among the points during the study period.

RESPONSE: We appreciate the concerns of the reviewer and recognize that the lack of precision in soil measurements is a limitation of this study. More precise measurements are not available and we felt that even though these measures were imperfect approximations of soil exposure, they could be indicative of risk. This approach was supported by similar methodologists that USGS has taken to produce continuous estimates of soil mineral concentrations throughout the contiguous United States [https://pubs.usgs.gov/of/2014/1082/pdf/ofr2014-1082.pdf]. 

Again, imprecision in measurement of exposure will likely lead to misclassification toward the null value, such that finding an association is likely indicative of a true effect. In two prior studies of NTM environmental risk, similarly imprecise measures of soil exposure were used but consistent associations were identified. For example, manganese was identified as a protective factor in two studies using various approaches in measurement (see Adjemian et al 2012, Lipner et al 2017 in main text references). Thus, although these studies and measurements have their limitations, they can indicate factors that may be causal and warrant further study. 

Regarding the kriging, we performed ordinary kriging on all soil mineral variables as well as each year for precipitation and temperature to obtain continuous estimates of the variables across the state of Florida. The average of each variable within each ZIP code was then used for analysis. 

This has been clarified in the manuscript: “Ordinary kriging was performed to estimate the broader spatial distribution over Florida from the original sampling sites or weather stations for all soil geochemistry variables, temperature, and rainfall.” 

When one looks at variability in soil characteristics, this reader was surprised by the extreme temporal and spatial variability in concentrations of minerals. Let’s examine the two that made it to the final model, sodium and manganese.

Sodium: the exposure analysis using grid spacing for sodium samples of 1 site per 1,600 km2 appears to be too coarse to estimate your subjects exposures at the zipcode level. See: Trangmar et al. Application of Geostatistics to Spatial Studies of Soil Properties. Advances in Agronomy 1986. https://doi.org/10.1016/S0065-2113(08)60673-2

Manganese: in a study of agricultural fields, manganese concentrations differed among months of the year and over the extent of the field. Areas with higher concentrations did not keep rank order of concentrations over the year. See: Hoskinson, et al. Temporal Changes in the Spatial Variability of Soil Nutrients. INEEL/CON-99-00290. 1999.

https://www.osti.gov/servlets/purl/9823-CNgTWm/webviewable/

QUESTION 9: It appears that soil minerals are variable spatially and temporally due to multiple factors such as land use, vegetative cover, soil microbial communities, and precipitation.

Therefore, I fear that your assignment of soil mineral exposure was too simplistic. Please prove me wrong, because we need to better understand environmental risk factors for NTM colonization.

RESPONSE: We recognize that soil mineral concentrations likely vary more than represented in our model and have clarified this as a limitation in the manuscript:

“Additionally, longitudinal, granular data on soil geochemistry is currently unavailable; the soil mineral concentrations throughout Florida likely vary more than represented in our study, limiting our ability to adjust these variables based on time and location.” 

When we average over time this may reduce some of the variability; it could be spurious, and that is why we feel that further investigation of these factors is warranted – we hope these data can serve as a call for further studies with more precise data. In addition, since many people are exposed to water, and soil mineral content is likely eventually reflected in the water, we feel that further studies which look at the water mineral content would be important.

---

## [Editor Report · Decision Letter 1]

2 Nov 2021

Environmental predictors of pulmonary nontuberculous mycobacteria (NTM) sputum positivity among persons with cystic fibrosis in the state of Florida

PONE-D-21-06131R1

Dear Dr. Ricotta,

We’re pleased to inform you that your manuscript has been judged scientifically suitable for publication and will be formally accepted for publication once it meets all outstanding technical requirements.

Kind regards,

Thomas Byrd

Academic Editor

PLOS ONE
---

## [Editor Report · Acceptance letter]

1 Dec 2021

PONE-D-21-06131R1 

Environmental predictors of pulmonary nontuberculous mycobacteria (NTM) sputum positivity among persons with cystic fibrosis in the state of Florida 

Dear Dr. Ricotta:

I'm pleased to inform you that your manuscript has been deemed suitable for publication in PLOS ONE. Congratulations! Your manuscript is now with our production department. 

Kind regards, 

on behalf of

Dr. Thomas Byrd 

Academic Editor

PLOS ONE